# Automated Detection of Presymptomatic Conditions in Spinocerebellar Ataxia Type 2 Using Monte Carlo Dropout and Deep Neural Network Techniques with Electrooculogram Signals

**DOI:** 10.3390/s20113032

**Published:** 2020-05-27

**Authors:** Catalin Stoean, Ruxandra Stoean, Miguel Atencia, Moloud Abdar, Luis Velázquez-Pérez, Abbas Khosravi, Saeid Nahavandi, U. Rajendra Acharya, Gonzalo Joya

**Affiliations:** 1Romanian Institute of Science and Technology, 400022 Cluj-Napoca, Romania; catalin.stoean@gmail.com; 2Department of Applied Mathematics, Universidad de Málaga, 29071 Málaga, Spain; matencia@ctima.uma.es; 3Institute for Intelligent Systems Research and Innovation (IISRI), Deakin University, Geelong 3216, Australia; m.abdar1987@gmail.com (M.A.); abbas.khosravi@deakin.edu.au (A.K.); saeid.nahavandi@deakin.edu.au (S.N.); 4Cuban Academy of Sciences, La Habana 10100, Cuba; cirahsca2@cristal.hlg.sld.cu; 5Center for Research and Rehabilitation of Hereditary Ataxias, Holguín 80100, Cuba; 6Department of Electronics and Computer Engineering, Ngee Ann Polytechnic, Clementi 599489, Singapore; aru@np.edu.sg; 7Department of Bioinformatics and Medical Engineering, Asia University, Taichung 41354, Taiwan; 8International Research Organization for Advanced Science and Technology (IROAST), Kumamoto University, Kumamoto 860-8555, Japan; 9Department of Electronic Technology, Universidad de Málaga, 29071 Málaga, Spain; gjoya@uma.es

**Keywords:** deep learning, medicine, sensor data, electrooculogram, uncertainty quantification, Monte Carlo dropout, decision trees

## Abstract

Application of deep learning (DL) to the field of healthcare is aiding clinicians to make an accurate diagnosis. DL provides reliable results for image processing and sensor interpretation problems most of the time. However, model uncertainty should also be thoroughly quantified. This paper therefore addresses the employment of Monte Carlo dropout within the DL structure to automatically discriminate presymptomatic signs of spinocerebellar ataxia type 2 in saccadic samples obtained from electrooculograms. The current work goes beyond the common incorporation of this special type of dropout into deep neural networks and uses the uncertainty derived from the validation samples to construct a decision tree at the register level of the patients. The decision tree built from the uncertainty estimates obtained a classification accuracy of 81.18% in automatically discriminating control, presymptomatic and sick classes. This paper proposes a novel method to address both uncertainty quantification and explainability to develop reliable healthcare support systems.

## 1. Introduction

Medical data modeling using deep learning (DL) has already been acknowledged as the next major advancement in computational support in complex and sometimes critical decision making tasks. Nevertheless, since this data is often ambiguous, model uncertainty also has to be taken into consideration.

The current paper addresses such ambiguities surrounding the assessment of electrooculogram (EOG) sensor data for an early recognition of presymptomatic behavior for spinocerebellar ataxia type 2 (SCA2) patients. In this particular task, the signals (saccades) that form the register of a labeled case may have characteristics pertaining to the neighboring classes. Moreover, the number of records in each register varies from person to person, depending on the physician’s consideration to continue testing until some confirmation of a certain diagnosis class is seen. Contrarily, there is no possibility of computational discrimination into presymptomatic-healthy or presymptomatic-sick classes solely based on signal shapes.

Machine learning has been employed to tackle this problem and obtain a proper discrimination into the three classes [1]. However, the performance has improved only when a supporting unsupervised technique (viz. self-organizing maps (SOM)) was used in parallel with DL [2] or to clean the data before the application of a deep neural network (DNN) [3]. In this context, uncertainty quantification (UQ) may be a promising mechanism to include in the DNN, thus allowing to better differentiate the presymptomatic condition by addressing both data ambiguity and model behavior.

The proposed methodology targets UQ by employing Monte Carlo dropout (MCD) within the DNN. Dropout has been used as an effective method to avoid overfitting in deep networks, and its Monte Carlo variant has been recently employed to capture the uncertainty in such learning models. It has been demonstrated to perform as a Bayesian approximation of a Gaussian process [4]. Therefore, many DL applications for medical image analysis and signal data have currently embraced MCD to measure model uncertainty. The MCD process assumes a number *N* of forward passes within the DL model. A different dropout 0-1 mask for neuron deactivation is used every time. Unlike regular dropout, MCD is applied both at training and validation/test stages, thus providing not one but N outputs in predicting the decision for each sample, which gives the possibility to compute statistical uncertainty estimates.

In addition to the standard integration of MCD with a DNN for atomic saccade labeling (as in the state of the art), the paper further analyzes the influence of uncertainty in the classification results at the higher register level. This means that, by referring to each validation register, several statistical measures (means, standard deviations) are collected for the label groups from the prior DL-MCD training at the saccadic signal level. Decision trees [5] (DT) are then used to learn the labeling mechanism for these statistical features pertaining to each register and output decision rules that go beyond similar saccadic shape.

The application of the DL-MCD-DT tandem for EOG data to diagnose SCA2 patients has yielded the accuracy of 81.18%. The findings suggest the importance of UQ within DL applications for medical data, as a means to handle both data and model uncertainty. In addition, the proposed system helps to provide a visual explainability of the computational model.

The paper is structured as follows. The description of the data set used for this work is given in Section 3. Recent works on the related use of MCD for addressing uncertainty in DL models for medicine (image and sensor data) are outlined in Section 2. A detailed outline of the inner mechanisms of using MCD for DL training, validation, and test phases are given in the first part of Section 4, however, with an emphasis on the particular conceptual issues surrounding the current medical application. The computation of the statistical quantifiers of uncertainty at the register level and the subsequent DT classification procedure based on the new features are described in the second part of Section 4. The experiments, results and discussion are depicted in Section 5.

## 2. Uncertainty Quantification in Deep Learning

Medical tasks have been modeled over the years by nature-inspired algorithms, especially artificial neural networks and the more recent and popular DL, that brought the much needed support in terms of time efficient diagnosis and clinical feature identification [6,7,8,9,10,11,12,13,14,15,16]. At the other end, data coming from different types of sensors have been also effectively analyzed by neural techniques [17,18,19,20,21,22]. At the intersection, medical data provided by sensors, such as ECG and EOG, have also been successfully tackled by DNN architectures that are able to handle temporal data [3,23,24].

However, blindly trusting the result (output) of such a model risks different undetected failures (e.g., removal of structures and most important features) [25]. As expressed by Tanno et al. [26] these predictive failures have two main reasons:The given task is naturally vague orThe models are not appropriate to describe the data.

These failures can be named uncertainty, and Malinin [27] has categorized them into two major groups:Aleatoric uncertainty or data uncertainty.Epistemic uncertainty or knowledge uncertainty.

To deal with uncertainty, different methods such as Bayesian methodology (e.g., MCD [28], variational inference [29]) and ensemble learning techniques [30] have been proposed in the literature. In this study we have used the MCD approach to capture the model uncertainty within DL. Gal and Ghahramani [4] proposed this Bayesian-based theoretical understanding of dropout as a sampling method which is equal to the variational approximation of a deep Gaussian process.

In the following, we have summarized some recent studies that employ MCD for UQ from DL in medicine. Generally, the emphasis is on medical image analysis as the most frequent DL application scenario, nevertheless there are a couple of entries also on MCD for DNN applied to medical sensor data.

Jungo et al. [31] found that uncertainty estimates are very important for different safety-critical computer-aided applications. Hence, they applied MCD in brain tumor cavity segmentation. The obtained results highlighted the significant potential of employing the extra information achieved from the model parameter uncertainty in validating the segmentation performance of various DL methods. In another study, Jungo et al. [32] investigated uncertainty measures of brain tumor image segmentation. According to the obtained outcomes, there is comparable segmentation performance between normal weight scaling dropout methods and MCD. Lubrano di Scandalea et al. [33] applied Deep and Active Learning (DAL) with U-Net architecture on histology data for the Axon-Myelin segmentation. They employed the MCD to appraise model uncertainty and choose samples to be used for the next iteration, which resulted in very efficient and straightforward results. Guo et al. [34] applied MCD in CNN (ConvNet/CNN) to engender a mean of U-net predictions. The use of MCD uncertainty improved the segmentation performance of the model applied to cardiovascular disease image data. Regarding specifically applications on medical sensor data, classification of electromyographic (EMG) hand gesture signals was done using three novel ConvNet methods by Côté-Allard et al. [35]. To deal with the overfitting issue, they employed the MCD, batch normalization and early stopping. In [36], the electromagnetic radiation effects of wireless devices on the brain are studied by using an autoencoder with MCD for UQ.

## 3. Materials

SCA2 is a neurodegenerative, hereditary disease that gradually affects movement. It is incurable; nonetheless, EOG makes the assessment of preliminary symptoms possible and consequently allows physicians to take action in decelerating the installation of the disease.

The data set used in this study was provided by the Center for Research and Rehabilitacion of Hereditary Ataxias (CIRAH), Holguín, Cuba. It can be downloaded for research purposes from the following link: https://dx.doi.org/10.6084/m9.figshare.11926812.

Eighty-five EOG tests with visual stimuli of four different angles were recorded. Segments from the velocity profiles were grouped into two categories using the *k*-means algorithm, and segments corresponding to the group with high velocity were considered as saccades. Each saccade has a length of 192 samples at regular time steps with normalized amplitude in the interval [−0.5,0.5] (see [2] for a detailed description of the acquisition and preprocessing stages). In total, there are 5953 saccades collected for all the individuals that had undergone examination. The saccades that belong to the same patient are held in a unique register. The register has one of the three labels: control (C), presymptomatic (P) and sick (S). The control class gathers the healthy eye movement, the sick contains the visibly deformed saccades due to the presence of SCA2 and the presymptomatic labels consist of the presumably precursory precondition.

Each register thus corresponds to one patient and has a unique class attributed to the three. It also consists of a variable number of saccades. For some cases the physician can rapidly reach the diagnosis, while for others several trials need to be performed to arrive at an accurate judgment. In this situation, a different number of saccades are produced. This underlines the fact that many of the samples in one patient register have a shape that does not necessarily reflect the pattern of the class attributed to that register, but there were a number of other saccades that made the physician arrive at that final decision.

To summarize, there are 85 registers (corresponding to the performed tests) and 5953 saccades in total, each with a length of 192. Figure 1 shows examples of saccades contained in registers of the three classes.

Figure 2 indicates the 40%–40%–20% split of the registers used for training, validation and testing.

The average, standard deviation (StD), minimum (Min) and maximum (Max) number of saccades present in a register for each class is presented in Table 1.

The difficulty of the current medical problem ensues from the different number and variable shape of saccades in each register, especially those with presymptomatic label. Hence, the differentiation is strongly impeded by the uncertainty in the data.

## 4. Methods

The proposed methodology comprises two stages: DL with MCD and DT. The former, depicted in Section 4.1, outlines the MCD approach at the training level of the DL model and the sample labeling at the validation/test stages, also on the premise of MCD. The latter, described in Section 4.2, consists in the construction of the DT using the feature information extracted from the MCD stochastic outcome for the validation registers and its prediction on the test registers with their MCD probabilistic label characteristics.

The methodological flow is depicted in Figure 3. The DL architecture with *N* MCD repetitions is appointed for the training data at the saccadic samples level. It outputs both the MCD ensemble accuracy and mean accuracy for validation and test sets over the *N* runs. Subsequently, the MCD application at the validation and test levels performs the labeling of individual saccadic samples in each register into the three classes. For each such formed group of saccades within a register, the cardinal, mean and standard deviations of the probabilities of constituent saccades belonging to each of the three classes are computed. The values for these new features are collected for every register and the ones corresponding to the validation set are fed to the DT, where they serve as training data. Hence, the validation samples have two roles: firstly, they are used to evaluate the CNN-LSTM model for individual classification of each saccade and, secondly, the aggregated results of the MCD on the validation registers are further used to train the DT model. Once the DT model is constructed, the decision rules are visualized and the classification accuracy is calculated for the test data using the statistical features.

### 4.1. Deep Modeling with Monte Carlo Dropout for EOG Saccades

The basic DL architecture chosen for the EOG data is a CNN-LSTM tandem that demonstrated to be most effective in the prior study [3]. The novel element in this work is the addition of MCD to achieve UQ.

The DL model starts with two convolutional layers with a kernel size of 3 and a number of filters of 128, ReLU activation and subsequent max pooling layers of size 2 and stride 2. The output is given to a LSTM layer with 100 units. At this point, in the current version of the architecture, MCD steps in with a dropout rate of 0.5. The final dense layer with 3 units and a softmax activation ends the sequence. An overview of the architecture is sketched in Figure 4.

Apart from its presence during the training stage, MCD appears also in the subsequent validation/test steps. Algorithms 1 and 2 depict how validation and test saccades are being labeled in the *N* consecutive MCD runs and in an ensemble fashion, respectively.

In Algorithm 1, during each run, the model is applied to every individual saccade. The probabilities of the current saccade belonging to each of the three classes is obtained (line 3). The maximum value of these three probabilities yields the prediction X∈{C,P,S} for the sample (line 4). The occurrence (0 or 1) of an accurate classification is computed as the indicator function of a match between the predicted label and the real one (line 5). The average classification accuracy at the end of the run is calculated (line 7) and the average over all the *N* MCD runs is obtained (line 9).
**Algorithm 1:** Calculation of class probabilities, labeling of individual saccades and computation of mean accuracy over the *N* runs of the MCD. This procedure is applied to both validation and test registers with constituent saccades
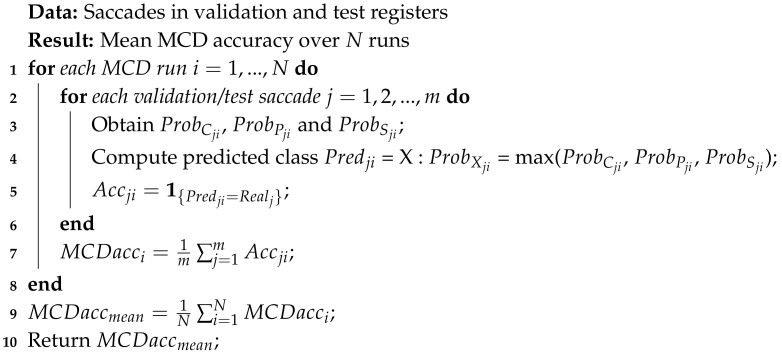


In Algorithm 2, for each saccade (validation or test), the model is applied over the *N* runs and its probability of belonging to each of the three classes is obtained for each run (line 3). Its mean probabilities for each class are then computed for the corresponding values obtained over *N* MCD runs (lines 5–7). The class predicted for the current saccade is the maximum value of the three mean probabilities (line 8). The prediction accuracy (0 or 1) for the current saccade is calculated (line 9) and the average accuracy of the ensemble over all saccades is reached (line 11).
**Algorithm 2:** MCD ensemble (MCDE) calculation of class probabilities and labeling for individual saccades and computation of accuracy. This procedure is applied to both validation and test registers with constituent saccades
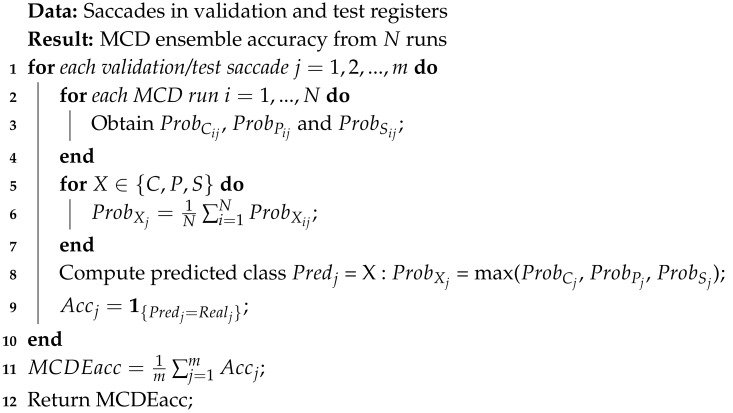


### 4.2. Feature Extraction and Classification of Registers via Decision Trees

Although the accuracy returned by the ensemble model from multiple MCD runs (MCDEacc in Algorithm 2) tops the accuracy of the single CNN-LSTM approach or the average over the MCD runs ($MCDacc_{mean}$ in Algorithm 1), it still does not solve the complex problem of classifying correctly the registers of saccades, specifically recognizing the presymptomatic behavior.

Algorithm 3 presents a mechanism to reach a label of registers, using new features which count the probabilities for each class of the constituent saccades and a DT approach to mine this novel information.

The classification at the register level starts from the previous Algorithm 2 that establishes the ensemble computation of class probabilities and labeling for every validation and test saccade. Therefore, the saccades from each register can be arranged in groups (line 3 from current Algorithm 3) following the labels that were predicted in line 8 of Algorithm 2. For every group, the mean and standard deviation of probabilities found in saccades by lines 5–7 of Algorithm 2 are calculated for every three classes (lines 5–10). Note that there may be obviously groups with no attributed saccades, e.g., for a certain register *i*, there may be no saccade labeled as presymptomatic so that $P_{pred}^{i}=\emptyset$. In this case the mean and standard deviations $Mean_{XY_{pred}}^{i},StD_{XY_{pred}}^{i}$ with X=P cannot be computed.

All these new features are recorded for every register: the number of saccades in each found label group and all the mean and standard deviation values from the probabilities for saccades also per assigned group. This results in 3 (for cardinals of each found group) + 9 (for means) + 9 (for standard deviations) = 21 features for each register.

A DT with a Gini metric is constructed from the features of the validation registers (line 12). Once the model is built, it is applied for the test registers (line 13) so that the classification accuracy as well as a visualization of the decision rules are obtained (line 14).
**Algorithm 3:** DT model construction on the validation registers and prediction on the test registers
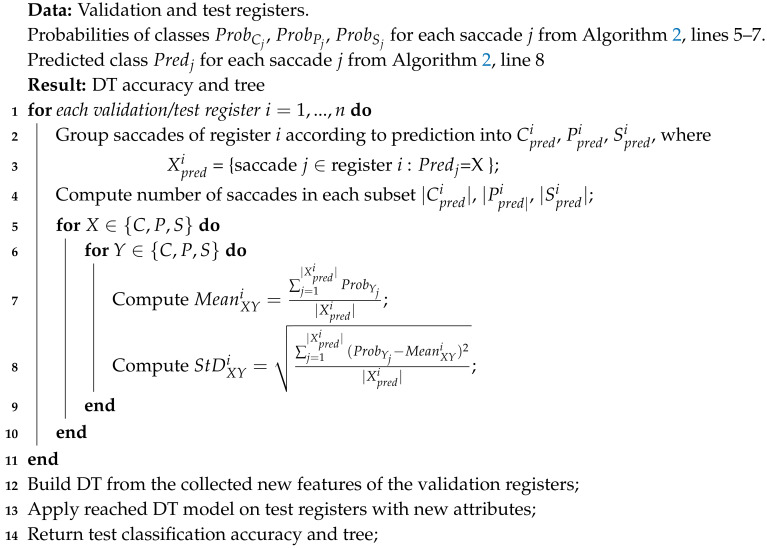


## 5. Experimental Results

The current section illustrates the results obtained by our proposed approach (in Algorithm 3) using the described data set. Section 5.1 discusses the configuration of the experiment, Section 5.2 illustrates the visual representation of results and the outputs are represented in Section 5.3.

### 5.1. Experimental Setup

Figure 2 shows the training, validation and testing registers used for this experiment. Due to the stochastic nature of the procedure, to gather statistically consistent data, there are 10 repeated runs on the same data splits for the entire DL approach using MCD with 500 passes. Accordingly, 10 slightly different data sets with features are created and finally the DT are built on each of these.

The mean accuracy result of DT is obtained after running the model 100 times. The large number of repeated runs is possible due to fast running time and hence it confirms the impartiality in the results.

### 5.2. Results and Visualization

Figure 5 shows the distribution of predictions over the 500 MCD runs (Algorithm 1) and the accuracy of the ensemble (Algorithm 2) in one of the 10 repetitions of the DL.

Figure 6 illustrates the manner in which the MCDE approach outputs the probabilities for each of the three classes (control, presymptomatic and sick) Algorithm 3, by showing one particular register in each class. The number of saccades included in each box plot is mentioned in the title of the corresponding chart.

The DT used a subset of features saved from the MCDE approach, as described in Section 4.2, to build a model that is able to classify the registers. The description shown in Figure 7 is the one that yielded the best test accuracy from one of the 10 repeated runs of the entire DL-MCDE-DT approach. More details about the interpretability of the DT rules are discussed in the next section.

Figure 8 shows how a presymptomatic register is mistaken as control by another DT.

Table 2 has two sections, one dedicated to the saccade classification results on the validation and test sets, as obtained by MCD and its ensemble version along the CNN-LSTM without Monte Carlo uncertainty and a support vector machine (SVM). All results are reported out of 10 repeated runs. The second part of the table shows the results obtained by the DL-MCDE-DT model, the CNN-LSTM without MCD and SVM, when applied to the register test set. For the last two approaches, the label of a register is established as the class of the majority of its saccades. The last 3 rows depict the precision, recall and F1-score obtained by the proposed DL-MCDE-DT.

Finally, Figure 9 indicates on the left plot the correctly and mislabeled registers in the 10 runs corresponding to Table 2 register accuracy results, while the right one shows the ROC curves in one of the 10 runs.

### 5.3. Discussion

When the class for each register is taken directly as the vote given by the majority of its constituent saccades, the results are not encouraging: the control and sick registers are correctly identified, but most of the presymptomatic registers are mistaken.

Figure 6 depicts the results of MCDE probabilities for three distinct registers, one representing each class. To further dissect the output, the attention is next focused on the plots of the first row. The register C020 from the control class has 85 saccades and 83 out of these are labeled by the MCDE approach correctly. This is depicted in the first plot that shows the probabilities for the saccades of register C020 that have the largest value for control. There is only one saccade that has a larger probability for the presymptomatic class (top-center plot) and one saccade where the largest probability targets the sick class (top-right plot). The majority of saccades (83 out of 85, that is 97.6%) are in this case labeled correctly (as control).

The second row of plots shown in Figure 6 illustrates a presymptomatic register. However, the MCDE does not classify any of the saccades as presymptomatic. Nevertheless, out of 52 saccades, 33 are classified as control (center-left plot) and 19 are labeled as sick (center-right plot). Hence presymptomatic saccades are misclassified. The decision on the label of the register needs to be established mainly by balancing the control and sick saccades.

Finally, the plots corresponding to the third row of Figure 6 correspond to the classification of the 56 saccades in the sick register S008. In this, 2 may be labeled as control, 1 as presymptomatic and 53 as sick. The ones labeled as sick are established by the DL-MCDE with a remarkably high certainty.

Naturally, the three registers from Figure 6 do not necessarily reflect the manner in which the saccades are labeled in all the other registers from the same corresponding classes. There are control registers in which all saccades are correctly identified in their entirety, while there are also others where more saccades are mislabeled. We initially attempted to manually establish rules (with thresholds) for balancing the control and sick saccades towards reaching an accurate classification of the validation registers. However, this path was abandoned as the rules became too complex to follow. Consequently, we extracted various statistical features at the register level from the obtained results and fed them to a DT model to extract the rules.

Figure 7 illustrates such a tree with rules obtained by the DT model. The most important attribute, i.e., the one from the root with the highest gini value, is represented by the number of saccades that are labeled as control. If there are less than 22 samples in the same register that are labeled as control by the MCDE model, the class of that register is established as sick. Figure 2 shows the overview with respect to the amount of saccades for each register and three classes. Most of the registers that have a limited number of saccades (e.g., less than 30) belong to the sick class (and the number of control saccades naturally falls below 22). When evaluating such a patient, the physicians decided that no more tests are necessary, since they observed an impaired behavior and this is accurately identified by the DL-MCDE approach, as well. For other registers with more samples, when the number of control saccades was below 22, the label was sick for all validation cases. Actually, it can also be observed in Figure 9 that this rule proved to be accurate for the test registers too, since all sick class registers are correctly classified.

When there are more than 21 samples classified as control in a register, the differentiation is to be made between presymptomatic and control. The next most important attribute is represented by the mean probability of the sick class for the samples that are classified as sick by the MCDE (the rightmost box plots with *S* in Figure 6). Naturally, this attribute is very important, since it represents the average probability returned by the softmax activation in the CNN-LSTM approach according to which the saccades should be labeled as sick. However, this does not directly decides the class of the register, but it leads to a further check on the mean probability of the presymptomatic label, also for the set of saccades when the sick probability is the highest. In the same Figure 6, this corresponds to the middle box plot (labeled with *P*) from the same rightmost plots. Finally, the mean probability of control saccades in the same set of samples labeled as sick (i.e., cases with the highest probability for sick) is another decision attribute. This corresponds to the box plot labeled with *C* in the same rightmost charts of Figure 6.

Figure 8 illustrates another tree that is similar in some nodes with the one from Figure 7. This new illustration is concentrated in pointing how a test register with the presymptomatic class is categorized as control. Each node shows a histogram with the registers in each class: the horizontal line contains the interval for the current attribute and the black triangle indicates the determined gini value. The features involved in the DT classification are written in orange at the bottom of the plot and also indicated with an orange triangle in each node of the tree. The run whose result is outlined in the figure had only one mistaken register in the test set, i.e., the one represented.

One drawback of the MCDE approach is given by the running time. While applying 500 passes of the MCDE over the validation set takes 24.16 min (2.9 s per iteration), the same amount of applications on the test set takes 11.33 min (1.36 s per iteration). We recall that the test set is smaller than the validation set, as shown in Figure 2. The experiments are performed on a PC with an Intel i7-4770 CPU, 3.40 GHz, 16 GM RAM and a GPU GeForce GTX 1650. The program is written in Python, uses the TensorFlow library and it runs on the GPU.

The DT in Figure 7 provided the highest classification accuracy of 94.12% for the test registers. It misclassified one presymptomatic register for one control. It is interesting to acknowledge that besides the first attribute that refers to the number of saccades labeled by the model as control, all the rest used only mean probabilities from the set of samples in the registers that have the highest probability for the sick class (corresponding to the box plots from the rightmost charts in Figure 6). Naturally, not all DT rules from the 10 repeated runs were identical and different attributes were also considered in other cases.

The results of saccade classification shown in Table 2 indicate the best result for the CNN-LSTM approach, both for the validation and the test sets. The high advantage is however not preserved when the classified samples are used to establish the label of the register. Despite the fact that all control and sick registers are accurately identified by taking the class of the majority of the saccades, the presymptomatic registers are all misclassified for control when the additional DT is not used.

The second part of the table indicates the results obtained for register classification. Besides the accuracy of the DL-MCDE-DT tandem, the CNN-LSTM and the SVM results are also reported, with the majority of samples establishing the label of their register. Although the values from the standard deviation in the saccade classification for CNN-LSTM indicate some spreading, this is not enough to change any label at the level of the register, hence the null value for the standard deviation in the second part of the table for the same classifier. Afterwards, the weighted results for the precision, recall and F1-score are shown. As it can be observed in the first plot from Figure 9 with the confusion matrix, all sick registers are correctly identified and no other register is mistaken for sick, as opposed to the output in [3]. The matrix is symmetric, hence there are very close values for the precision, recall and F1-score in Table 2. The values for the three measures in the table are not however identical to those in the figure, because they are computed as average over all 10 runs and not directly from the confusion matrix results.

A higher degree of presymptomatic signs are now correctly identified, as opposed to the results in [3]. This is also visible from the right plot in Figure 9 with the ROC curves, which are calculated for one of the 10 runs. The micro- and macroaverages are also computed. The high value for the microaverage is of special interest, since the classes of the problem are unbalanced (more control registers and less presymptomatic ones) and this measure adequately captures the precision in such cases. It would still be useful to have a significantly larger number of registers to train the DT model with more data and make it more robust.

## 6. Conclusions

This paper investigates the need and benefits of UQ in the medical field where the main intricacy comes from the uncommon nature of the sensor-derived data. Our proposed model is able to classify the person suspected of SCA2 in one of the three classes: control, presymptomatic and sick. However, the register of an individual is formed by several saccadic movements obtained from the electrooculogram and the class is defined based on the entire set. The complexity results from the fact that the presymptomatic registers contain many samples that have the normal behavior, representative for control and also some that are typical for sick.

A CNN-LSTM model using Monte Carlo dropout learned to distinguish between the validation saccades. In each validation and test register, subsets of saccades are classified as control, presymptomatic and/or sick. For each of these subsets, mean and standard deviation are computed from their probabilities. The new numerical validation data set is fed to the DT for training, with an aim to better assess the classes of the registers. The rules of the DT are further applied on the test data with the same type of features.

Our proposed model obtained an average overall classification accuracy of 81.18% (as opposed to the previous 78.24% reported in [3]), with a relatively good identification of presymptomatic registers. Moreover, it also provides explainable rules for the decision.

The use of UQ can be further explored for the current problem by paying special attention to the saccades where the uncertainty is very low or, on the contrary, to those where its value is very high. These numbers could be used as features that might determine the type of the register and are considered as a future line of research.

## Figures and Tables

**Figure 1 sensors-20-03032-f001:**
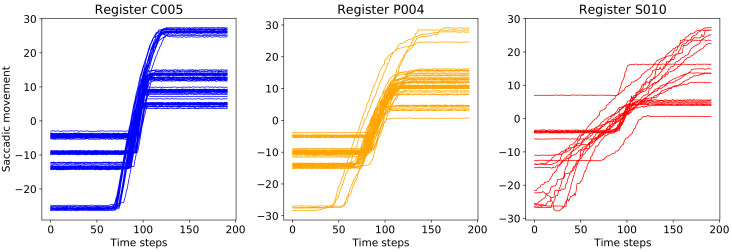
Examples of 3 random registers used for training: from **left** to **right**, for control, presymptomatic and sick.

**Figure 2 sensors-20-03032-f002:**
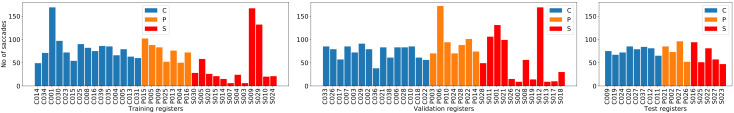
Each plot shows the number of saccades for each register in turn. From **left** to **right**, the registers considered for the training, validation and test sets are shown.

**Figure 3 sensors-20-03032-f003:**
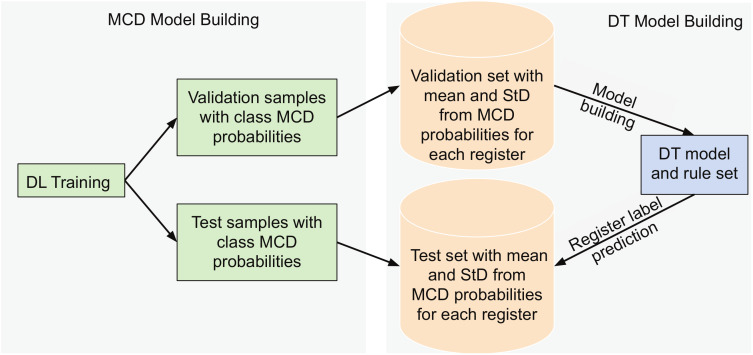
Flowchart of the proposed DL-MCD-DT learning: DL training of saccades with MCD, statistical feature extraction from groups in validation registers, DT model construction and classification and rules on the test registers.

**Figure 4 sensors-20-03032-f004:**
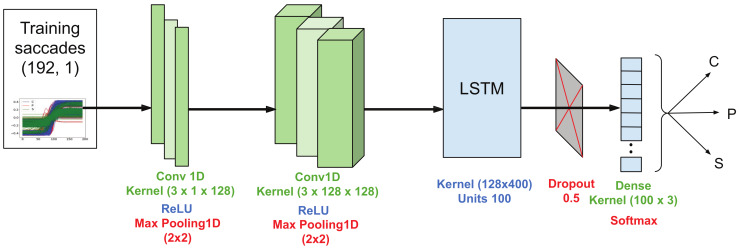
Proposed CNN-LSTM architecture. MCD intervenes in the final dropout layer, with a different mask for every forward pass.

**Figure 5 sensors-20-03032-f005:**
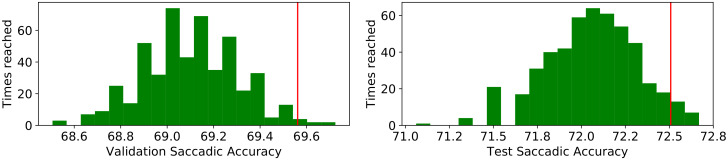
Distribution of the 500 MCD accuracy predictions ($MCDacc_{i}$ in Algorithm 1) for the validation (**left**) and test (**right**) sets in a single run of the approach. The vertical line represents the classification accuracy of the ensemble MCD (MCDE in Algorithm 2) model.

**Figure 6 sensors-20-03032-f006:**
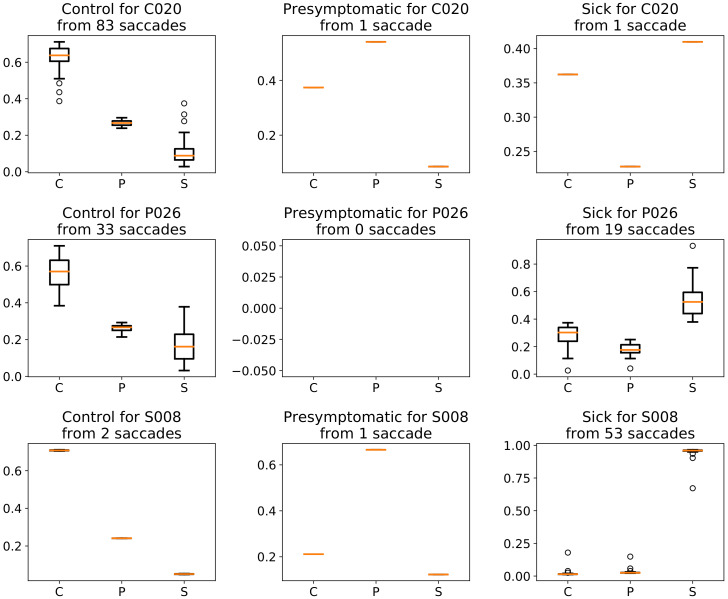
Box plots of 3 registers corresponding to each class of the problem: first row corresponds to a control register, second refers to a presymptomatic register and third to a sick register. For each register, mean and standard deviation results are outlined for saccades with the labels given by the MCDE approach, as described in Algorithm 3.

**Figure 7 sensors-20-03032-f007:**
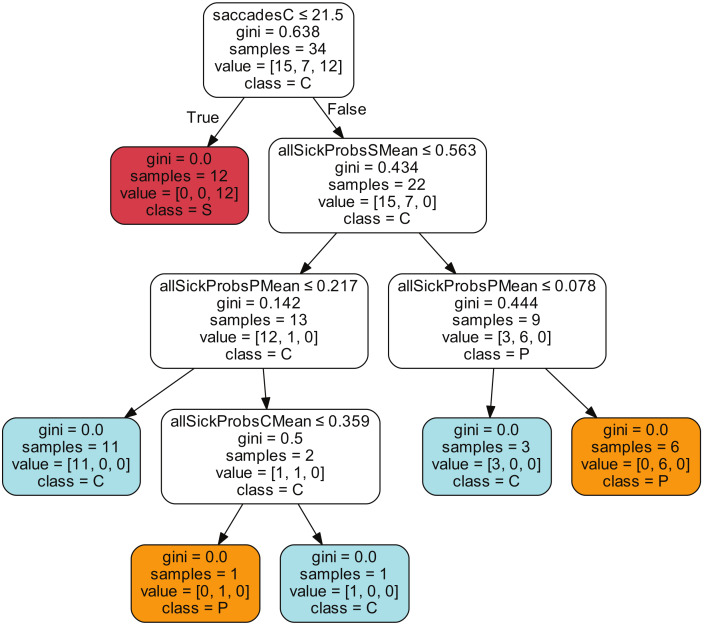
Illustration of the decision tree model obtained from the validation data in one of the repeated runs.

**Figure 8 sensors-20-03032-f008:**
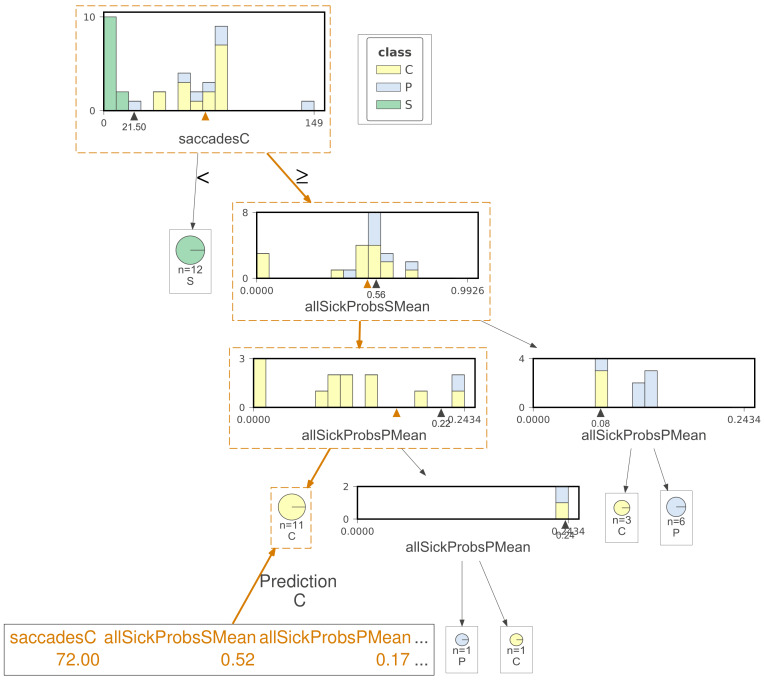
One presymptomatic register wrongly classified as control by a decision tree. The features involved in the decision appear at the bottom of the plot.

**Figure 9 sensors-20-03032-f009:**
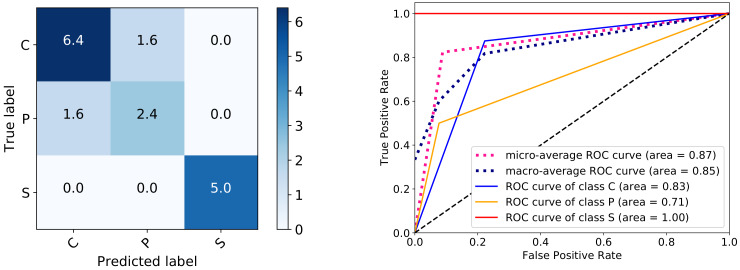
Confusion matrix of test registers obtained after 10 repeated runs of the DL-MCDE-DT model (**left**) and ROC curves for one run (accuracy 82.35%) (**right**).

**Table 1 sensors-20-03032-t001:** Average, standard deviation, minimum and maximum number of saccades per register in the entire data set.

Class	Average	StD	Min	Max
Control	78.3	21.6	38	169
Presymptomatic	76.7	25	28	172
Sick	54	49	6	172

**Table 2 sensors-20-03032-t002:** Summary of classification performance obtained for various combinations of approaches and for other methods.

Approach	Average (%)	Standard Deviation (%)	Minimum (%)	Maximum (%)
Saccade classification				
DL-MCD validation	68.07	0.74	66.68	69.29
DL-MCD Ensemble validation	68.77	0.51	67.78	69.48
CNN-LSTM validation	70.38	0.28	69.7	70.7
SVM validation	60.06	0	60.06	60.06
DL-MCD test	70.28	1.15	68.57	72.19
DL-MCD Ensemble test	70.84	1.29	68.81	72.75
CNN-LSTM test	74.12	0.17	73.9	74.4
SVM test	63.18	0	63.18	63.18
Register classification on the test set				
DL-MCDE-DT Accuracy	81.18	7.16	72.12	94.12
CNN-LSTM Accuracy	76.47	0	76.47	76.47
SVM Accuracy	64.71	0	64.71	64.71
DL-MCDE-DT Precision	81.9	9.05	68.63	94.77
DL-MCDE-DT Recall	81.18	9.04	70.59	94.12
DL-MCDE-DT F1-score	80.93	9.16	69.36	93.87

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
