# Peer review of "Automated Detection of Presymptomatic Conditions in Spinocerebellar Ataxia Type 2 Using Monte Carlo Dropout and Deep Neural Network Techniques with Electrooculogram Signals"

_sensors, 2020, doi:10.3390/s20113032_

Round 1
Reviewer 1 Report
This manuscript studies uncertainty in a DL model in the field of healthcare. The MCD is applied to the CNN-LSTM structure to estimate the uncertainty of the model. In addition, DT is used to learn the labeling mechanism. It is written and structured well. However, it can be improved. My comments are as follows: 1. DT is used to learn the labeling mechanism, but not mentioned in the abstract. 2. The introduction is not written-well and lack of related information. 3. In section 2 some paragraphs are short and some are long. 4. Check the format of algorithms. 5. In Section 5, the performance of the proposed method can be compared with other methods either deep learning ones or traditional machine learning algorithms. In addition, it is better to compare the results of the proposed method with the original CNN-LSTM, then readers can see the improvements.
Reviewer 2 Report
This paper investigates the need and benefifits of UQ in the medical fifield where the main intricacy comes from the uncommon nature of the sensor-derived data. Our proposed model is able to classify the person suspected of SCA2 in one of the three classes: control, presymptomatic, and sick. This topic is interesting and the paper is overall well organized. Following are some major concerns:
- The detailed architecture diagram of the proposed network should be given.
- The dataset used in this paper should be explained in details. And link should be given if it is an open dataset.
- More comparisons should be done other than classification accuracy.
- What is the main aim of section 2? It should be re-organized.
Reviewer 3 Report
The presented method in the paper utilizes uncertainty information from Monte Carlo dropout and extracts features of uncertainty information for accurate classification of disease state in electrooculogram. The paper is well written. I have a few comments.
- page 5, line 155: "a CNN-LSTM tandem that demonstrated to be most effective in the prior study [3]". => Please provide a figure that describes the architecture of CNN-LSTM model. The performance of the CNN-LSTM was not demonstrated in the paper.
- Ref [3] "Stoean et al., A Hybrid Unsupervised ..." may have the details on the performance of the model, but it has not been published yet.
- Section 2 "Uncertainty Quantification: Monte Carlo dropout" introduces many previous literatures regarding Monte Carlo dropout, but some of them do not seem related to the paper. The content can be reduced without loss of the gist of the paper.
- In my understanding, validation samples seem to have two roles. First, they are used to evaluate the CNN-LSTM model for individual classification of each saccade. Second, they are used to train the decision tree model. It may be better to clarify this in the Methods section.
- line 199, 200: perhaps typos. line 15 -> line 13, line 16 -> line 14.
- line 207: "10 repeated runs". It is not clear whether the authors mean a 10-fold cross-validation. Or is it repeated after a random assignment of (C, P, S) grouping?
- Computation time may be a drawback in the Monte Carlo dropout method. Please provide mean and standard deviation of computation time for the DL-MCDE-DT classification per subject, with comparisons.
- Figure 7: It may be good to show example cases for the misclassification. Perhaps, it may be good to add a row to Figure 5 that describes the case of (True label=P, Predicted label=C) or otherwise.
- It seems there exists one ground truth reference. How is it reliable? I'm wondering if there is any issue of interobserver variability.
